# Alternatives to Carbon Dioxide in Two Phases for the Improvement of Broiler Chickens’ Welfare during Stunning

**DOI:** 10.3390/ani14030486

**Published:** 2024-02-01

**Authors:** Daniel Santiago Rucinque, Antonio Velarde, Aida Xercavins, Aranzazu Varvaró-Porter, Troy John Gibson, Virginie Michel, Alexandra Contreras-Jodar

**Affiliations:** 1Animal Welfare Program, Institute of Agrifood Research and Technology (IRTA), 17121 Monells, Spain; santiago.rucinque@irta.cat (D.S.R.); antonio.velarde@irta.cat (A.V.); aida.xercavins@irta.cat (A.X.); aranzazu.varvaro@irta.cat (A.V.-P.); 2Department of Pathobiology and Population Sciences, Royal Veterinary College, University of London, Hatfield AL9 7TA, UK; tgibson@rvc.ac.uk; 3Direction of Strategy and Programmes, French Agency for Food, Environmental and Occupational Health & Safety (ANSES), 14 Rue Pierre et Marie Curie, 94701 Maisons-Alfort, France; virginie.michel@anses.fr

**Keywords:** controlled atmosphere stunning, gas stunning, carbon dioxide, inert gases, nitrogen, unconsciousness, death, aversion, broiler chicken

## Abstract

**Simple Summary:**

Stunning during the slaughter process consists of inducing unconsciousness in animals in order to prevent them from feeling any avoidable pain, distress, or suffering during bleeding and related operations. In an unconscious state, the animal is unable to perceive and respond to any external stimuli, including pain. Currently, the two main stunning methods used commercially in broiler chickens are the electrical waterbath and carbon dioxide in two phases. Although the latter is widely recommended over electrical waterbath stunning, it is still not devoid of risks for animal welfare. For instance, the induction to unconsciousness is not immediate and involves a transitional period during which aversion to the inhalation of carbon dioxide might occur. The present study demonstrates that mixtures of carbon dioxide with nitrogen improve a broiler chicken’s welfare during stunning since they result in a more rapid induction of unconsciousness and reduce the aversion experienced, compared to carbon dioxide in two phases, in broiler chickens.

**Abstract:**

This study evaluated the exposure to gas mixtures of carbon dioxide (CO_2_) associated with nitrogen (N_2_) as alternatives to CO_2_ in two phases to improve the welfare of broiler chickens at slaughter. Broilers were exposed to one of three treatments: 40C90C (1st phase: <40% CO_2_ for 2 min; 2nd phase: >90% CO_2_ and <2% O_2_ for 2 min, *n =* 92), 40C60N (40% CO_2_, 60% N_2_, and <2% O_2_ for 4 min, *n =* 79), or 20C80N (20% CO_2_, 80% N_2_, and <2% O_2_ for 4 min, *n =* 72). Brain activity (EEG) was assessed to determine the onset of loss of consciousness (LOC) and death. Behavioural assessment allowed for characterisation of an aversive response to the treatments and confirmed loss of posture (LOP) and motionlessness as behavioural proxies of LOC and brain death in 40C60N and 20N80C. However, the lack of quality of the EEG traces obtained in 40C90C did not allow us to determine the onset of LOC and brain death for this treatment. The onset of LOC in 40C60N was found at 19 s [14–30 s] and in 20C80N at 21 s [16–37 s], whereas a LOP was seen at 53 s [26–156 s] in 40C90C. Birds showed brain death in 40C60N at 64 s [43–108 s] and in 20C80N at 70 s [45–88 s]), while they became motionless in 40C90C at 177 s [89–212 s]. The 40C90C birds not only experienced more events of aversive behaviours related to mucosal irritation, dyspnoea, and breathlessness during induction to unconsciousness but were at risk of remaining conscious when the CO_2_ concentration was increased in the 2nd phase (known to cause severe pain). From an animal welfare point of view, 40C60N proved to be the least aversive of the three treatments tested, followed by 20C80N and 40C90C.

## 1. Introduction

Pre-slaughter stunning is mandatory in the European Union [1] and many other countries worldwide [2]. It consists of inducing unconsciousness in animals in order to prevent them from feeling any avoidable pain, distress, or suffering during bleeding and related operations. In an unconscious state, the animal is unable to perceive and respond to any external stimuli, including pain [3]. To protect animal welfare at slaughter, it is essential to induce fast and effective stunning for enough time so that the animals do not regain consciousness before brain death due to bleeding.

Currently, the two main stunning methods used commercially in broiler chickens are electrical waterbath stunning (WBS) and controlled atmosphere stunning (CAS). WBS entails live bird shackling; pre-stun shocks may occur, and the stunning is not always effective. Therefore, a great concern regarding the bird’s welfare exists [4]. For this reason, CAS emerged as an alternative stunning method to WBS [5]. It consists of exposing large numbers of broiler chickens to modified atmosphere environments (e.g., carbon dioxide in two phases) or reducing the atmospheric pressure (LAPS), which induces a gradual loss of consciousness (LOC), and if the duration of the exposure is long enough, it causes death. However, CAS methods are not devoid of risks for animal welfare. For instance, the induction of unconsciousness is not immediate and involves a transitional period, during which negative welfare outcomes may occur [6].

Although WBS is the most common method used in the European Union, the number of slaughterhouses using CAS has dramatically increased during the last few years [7,8]. Most slaughterhouses with CAS use carbon dioxide (CO_2_) in two phases, while very few use CO_2_ associated with inert gases, and none of them use either inert gases or LAPS at present [8].

Commercial CAS equipment for poultry differs in design, as it is either tunnels, pits, or closed cabinets. In tunnels and pits, birds enter the system in their transport crates or they are uncrated by tilting the container [4] and enter the system on a conveyor belt. In tunnels and pits, the system is pre-filled with gas and birds enter continuously at one end of the system, and, while they are conveyed to the opposite end, they are exposed to different gas concentrations. In closed cabinets, birds enter the system in their transport crates, one batch at a time. Once the birds have been loaded into the system the gas is then added, and it is removed upon completion of the stunning cycle [8].

The physiological principle during the induction of unconsciousness using CO_2_ is to induce acidosis and neuronal depression [9]. However, prior to LOC, CO_2_ activates chemoreceptors in the mucous membranes of the respiratory tract, which may induce discomfort, pain, and breathlessness, as shown by behaviours indicative of aversion [10]. The degree of aversion depends on the CO_2_ concentration. The higher the concentration, the greater the aversion, but the more rapid the LOC [11]. To reduce aversion during the induction of unconsciousness, stunning with CO_2_ in two phases is carried out. It consists of exposing broiler chickens to an initial concentration of up to 40% CO_2_ until LOC occurs. Thereafter, the CO_2_ concentration is increased in the second phase, inducing a deeper state of unconsciousness and then death while birds are unconscious [4].

As an alternative to CO_2_ in two phases, the exposure to inert gases (e.g., nitrogen N_2_; argon Ar) is expected to reduce aversion. Inert gases are colourless, odourless, tasteless, and non-irritative, and, therefore, it is presumed that they are imperceptible to birds due to the lack of chemoreceptors in their air ways [5,12]. In addition, inert gases displace oxygen (O_2_) from atmospheric air, and this ensures that the birds lose consciousness by anoxia [3]. In this sense, the inhalation of inert gases is expected not to cause aversive reactions [13,14]. When birds enter a chamber filled with inert gases, their behaviour does not differ from when they are breathing atmospheric air [13]; they do not withdraw [11] and they barely show behavioural signs of distress [10,15]. Nevertheless, when birds are unconscious, they present severe convulsion expressed as wing flapping [10,13], which may cause self-inflicted injuries (wing fractures) or injuries and pain to the other birds that have not yet lost consciousness [5,16]. However, it is not entirely clear whether the onset of wing flapping is a reflexive reaction occurring after the bird loses consciousness or whether the birds are still conscious and trying to escape from this modified atmosphere [16,17].

The European Food Safety Authority (EFSA) has pointed out that research evaluating stunning methods requires well-controlled studies under laboratory conditions to characterise the animals’ responses to the stunning method (i.e., the onset of unconsciousness and death, the magnitude of aversion) [18]. Measuring electrical brain activity by means of electroencephalography (EEG) is the most accurate method to assess the onset and the duration of unconsciousness and time to death [18], and therefore to discern between aversive behaviours that occur before LOC and likely unconscious convulsions. The aim of this study was to assess different gas mixtures of CO_2_ and N_2_ as potential alternatives to the exposure to CO_2_ in two phases for the improvement of animal welfare during the stunning of broiler chickens. To this end, we aimed to compare EEG and behavioural recordings to determine the time to the onset of unconsciousness and death and to characterize the gas aversion response.

## 2. Materials and Methods

### 2.1. Experimental Design and Facilities

A total of two hundred and forty-three 39-day-old mixed-sex Ross 308 broiler chickens were transported from a commercial farm to the experimental facilities of the Institute of Research and Technology for Agriculture and Food (IRTA) in Monells (Spain). Birds were weighed (2.42 ± 0.18 kg) and individually identified with numbered leg bands before being allocated randomly to the different treatments. On arrival and after checking their health status and appropriate locomotor behaviour, birds were distributed randomly into seven adjacent lairage pens of 2 m × 1.8 m (35 broiler chickens per pen; stocking density of 23.5 kg/m^2^). Each pen was provided with litter material (wood shavings) and feed and water ad libitum throughout the experiment. The pens served as a lairage before slaughter but were not associated with any specific treatment.

The study was carried out at the experimental slaughterhouse of IRTA, located next to the lairage pens. It is equipped with a Dip-lift XL G2 gas stunning system (Butina Aps, Copenhagen, DK) that contains a lift (240 cm × 111 cm × 100 cm) with a perforated floor. The lift descended into the base of a pit at a depth of 290 cm. The pit was pre-filled with gas mixtures through an inlet valve placed at the bottom of the pit. CO_2_ and O_2_ concentrations were measured through a portable infrared single beam sensor for CO_2_ and an electrochemical sensor for O_2_ (Dansensor^®^ CheckPoint 3 O_2_/CO_2_, MOCON Europe A/S, Ringsted, Danmark) using one fixed sounding line placed at a depth of 260 cm and another mobile sounding line to check CO_2_ concentrations at different depths.

The experimental study lasted 5 days. On the first day, a subset of broiler chickens was exposed to atmospheric air (AIR, *n =* 100), serving as the control. These birds were reallocated in similar numbers to one of the three experimental gas treatments. Therefore, from d 2 to d 5, broiler chickens were stunned with one of the following gas treatments: CO_2_ in two phases, the 1st phase with <40% CO_2_ by volume in air for 2 min followed by the 2nd phase with >90% CO_2_ for 2 min (40C90C; *n =* 92); a gas mixture of 40% CO2 and 60% N_2_ with less than 2% residual O_2_ for 4 min (40C60N; *n =* 79); and gas mixture of 20% CO_2_ and 80% N_2_ with less than 2% residual O_2_ for 4 min (20C80N; *n =* 72). Each day consisted of two sessions: first session from 800 to 1200 h and second session from 1500 to 1900 h, with treatments alternating per session to avoid potential bias (Table 1). Each session consisted of 8 to 11 cycles (dips into the pit). In each cycle, four broiler chickens were placed in the lift and exposed to the treatment. In AIR, the behaviour of the four chickens per cycle was assessed, but in 40C90C, 40C60N, and 20C80N, in one of the four chickens, only the brain activity was assessed, while the other three were used for behavioural assessment (Figure 1). The schedule of the gas stunning treatments applied along the 5 d experimental period to broiler chickens, the number of birds used per cycle, the number of cycles, and birds used for EEG and behavioural assessments per treatment and session are summarised in Table 1. First session dedicated to 40C90C failed at registering the EEG, so an extra session with 40C90C was included at the end of the experimental period (see Table 1).

The bird used for brain activity assessment was placed on the lift and separated from its three other conspecifics by a transparent methacrylate wall with a floor area of 48 cm × 112 cm (0.53 m^2^). The separation was intended to prevent any disturbance to the birds and EEG signal interference from other birds. The three birds monitored for their behavioural activity were placed in the lift with a floor area of 144 cm × 112 cm (1.6 m^2^).

The exposure time was considered as being from when the lift started to descend into the pit until the lift arrived at its original position. The duration of exposure to each experimental treatment was determined from pre-trials aimed at assuring death in all birds at the end of the process. Gases used were pure CO_2_ and premixed mixtures of CO_2_ with N_2_ (Freshline gases^®^ for food use, Carburos Metálicos, Barcelona, Spain).

### 2.2. Gas Concentration Assessment

The pit was gas-filled before the birds entered, and CO_2_ and O_2_ concentrations were continuously monitored before, during, and after each cycle in all treatments. Gas concentrations were measured with a portable gas analyser (Dansensor^®^ CheckPoint 3 O_2_/CO_2_, MOCON Europe A/S, DK) every 10 cm vertically from the bottom to top of the pit. This allowed measures of CO_2_ and O_2_ concentrations at different depths.

In the 1st phase of 40C90C, the CO_2_ concentration varied, but was close to 40% and never exceeded that level throughout the cycles. Then, the distance from the top of the pit was registered and the lift descended until reaching that depth (53.1 ± 15.7 cm) to ensure that birds were exposed to the target concentration. During the second phase in 40C90C, and the cycles of 40C60N and 20C80N, the sounding line monitored the gas concentration at 30 cm from the bottom of the pit in order to monitor the gas concentrations at the level of the chickens’ heads.

### 2.3. Brain Activity Assessment

Fifty broiler chickens were randomly selected for electrical brain activity assessment through electroencephalography (EEG). Chickens were distributed into the 40C90C (*n =* 16), 40C60N (*n =* 16), and 20C80N (*n =* 18) treatment groups. 

Each bird was prepared prior the exposure to the gas treatment. Firstly, the bird was wrapped with a textile mesh to restrain it wings, body, and legs, leaving the head and neck exposed in order to minimize movement during the EEG recording. Secondly, the chicken’s neck was restrained gently (Figure 1). Then, head and neck feathers were shaved with an electric shaver and a gauze pad with topical alcohol was rubbed on the bare skin before subcutaneous administration of local anaesthesia to the head and neck. Local anaesthesia consisted in 0.1 mL of lidocaine 2% subcutaneously injected with an insulin needle and syringe in regions where EEG electrodes were to be placed. Once the skin was desensitized, three 24-gauge stainless steel subdermal needle electrodes (Neuroline Subdermal, Ambu Inc., Glen Burnie, MD, USA) were placed on the head as described in Gibson et al. [19]. 

The active electrode was inserted ≈6 mm right of midline and ≈3 mm cranial from the end of the comb over the right optic lobe. The reference electrode was inserted over the right rostral aspect of the forebrain, ≈6 mm right of midline and ≈20 mm caudal from the end of comb, and the ground electrode was inserted in the neck in the caudal direction (Figure 2). Electrodes were secured in position with surgical tape (Durapore, 3M, Maplewood, MN, USA). Inter-electrode impedance was established to be between 1.2 and 2.0 kΩ (MkIII Checktrode, UFI, Morro Bay, CA, USA), and electrode leads were further secured with a loose band of surgical auto-fixing tape around the neck (Coeban, 3M).

EEG signals were amplified and filtered with an analogue filter (dual Bio Amp, ADInstruments Ltd., Sydney, Australia) with low- and high-pass filters of 100 and 0.1 Hz, respectively. The analogue signals were digitalized (1 kHz) with a 4/20 PowerLab (ADInstruments Ltd., Sydney, Australia) converter and recorded using a laptop for offline analyses. Pre-treatment EEG signals were collected for 90 s while the bird was on the floor of the lift with the other three birds prior to their descent into the pit to obtain the normal EEG data (i.e., baseline) to compare with post-treatment results. EEG recordings were monitored, saved, and pre-processed using LabChart 8 Pro (v.8.1.21, AD Instruments, Dunedin, New Zealand) [19].

Spectral analysis of EEG recordings was used for detecting waveform changes that indicate the onset of unconsciousness. Spectral variables, including total power (Ptot), median frequency (F50), and spectral edge frequency (F95) were computed from the EEG data. Ptot represents the overall area under the power spectrum curve, F50 corresponds to the median frequency of the power spectrum curve, and F95 indicates the frequency at which 95% of the power spectrum curve is located [20]. On the other hand, the brain’s electrical activity recorded in the EEG was categorized into different frequency bands: Delta (<4 Hz), Theta (4 to 8 Hz), Alpha (8 to 13 Hz), Beta (13 to 32 Hz), and Gamma (32 to 200 Hz). 

In a conscious animal, the Alpha and Beta frequency bands predominate the EEG spectrum. Thus, a decrease in F50 and an increase in Delta frequencies suggest a transition from consciousness to unconsciousness [14,16,21], and, therefore, the change in the relative contributions of F50 and Delta frequencies in the spectrum was used to estimate the loss of consciousness (LOC). The Ptot represents the total power on the spectrum, and a decrease in Ptot is associated with reduced EEG activity. This reduction was observed in the EEG of unconscious broiler chickens by Raj et al. [21] and Sandercock et al. [22], hence, the reduction in Ptot was employed to estimate the onset of death during spectral analysis [14,15]. The isoelectric pattern was observed on the filtered trace as a visual indicator of brain death, as in other studies [14,16,19,23,24]. 

### 2.4. Behavioural Assessment

Broiler chickens’ behaviour during exposure to the gases was recorded using three video cameras (IP Camera DH-IPC-HDW2231TP-ZS-S2, Zhejiang Dahua Vision Technology Co., Ltd., Hangzhou, China) and a digital voice recorder (Olympus VN-712PC, Olympus imaging Corp., Tokio, Japan). Two video cameras were placed inside the lift on each of the laterals and one in the central part of the lift’s ceiling. Video cameras were connected to a digital image recorder (Network video recorder DHI-NVR4108-8P-4KS2/L, Zhejiang Dahua Vision Technology Co., Ltd., Hangzhou, China). Then, the video records and audios were synchronized. 

These records helped in assessing the birds’ behaviour retrospectively, used by an assessor blinded to the experimental treatments. Behavioural observations were assessed continuously at the individual level; each bird was observed for 4 min (i.e., from the time the lift started to descend into the pit until the end of gas exposure) using BORIS (Behavioural Observation Research Interactive Software) v.7.13.8 [25] based on the ethogram shown in Table 2.

Loss of posture (LOP) was considered a behavioural indicator of the onset of unconsciousness [10,26,31,32]; therefore, behaviours appearing before LOP were considered voluntary behaviours related to aversion (e.g., pain, distress, breathlessness) as the birds were still conscious during gas exposure, while behaviours appearing after LOP were considered related to convulsions or any other involuntary movements. For this reason, behaviours were separated and annotated into two different groups: those occurring before and after LOP. Motionless was considered the behavioural indicator of brain death [24,27,28,30].

### 2.5. Data Pre-Processing and Statistical Analyses

EEG recordings were pre-processed using LabChart 8 Pro (v.8.1.21, AD Instruments, Dunedin, NZ). First, EEG recordings were digitally filtered to remove noise interference (band pass: 1 to 30 Hz). Then, epochs of 1 s, from baseline and during the 4 min of gas exposure, were selected for spectral analysis. Data were set at 1K Fast Fourier transformed, Hamming windowed, 50% window overlap, and zero frequency removed. For each bird, the following spectral data variables were calculated from each epoch: total power (Ptot, µV2), spectral median frequency (F50, Hz), and contribution of Delta frequency (1–4 Hz) to Ptot (%). The median value from the baseline was calculated for all variables at the individual level. Data generated were exported and analysed using Microsoft Excel 2016 (Microsoft Corporation, Redmond, WA, USA). Then, F50 values under 4 Hz on the baseline were removed in order to discard low-frequency artefacts caused by bird movements. 

Onset of LOC was calculated as the mean time at which F50 decreased below 50% [24,27,28] and the Delta frequency increased above 65% in comparison to the baseline value in four consecutive epochs. Brain death was determined via spectral analysis to be when Ptot decreased by 90% in comparison to the baseline values in four consecutive epochs [21,27] and visually when the trace was isoelectric, representing an almost flat line with very low Ptot (<2.5 µV). This is a pattern on an EEG related to a permanent state of unconsciousness or brain death [19,24,27,28,30,33,34]. Spectral and visual analyses have been well-established in previous studies [27,28,33,35,36,37], allowing for a more accurate estimation of the time to death and its relationship with behaviour.

At the end, no EEG trace from 40C90C could be used due to the low quality of the records. Time to LOC and brain death fulfilled the normality and homoscedasticity conditions in the remaining gas treatments (i.e., 40C60N and 20C80N). Therefore, Student’s *t*-test was used for the comparison of the two means. 

Behavioural data pre-processing, statistical analyses, and plots were performed using R software v.4.3.2. [38]. For all the statistical analyses, significance was declared at *p* < 0.05.

The analysis of behavioural measurements comprised the time to onset of LOP and motionlessness, the proportion of broilers that performed the rest of the behaviours listed in the ethogram, as well as the number of events per bird, and total duration of each, both before and after LOP, per treatment. In order to avoid potential pseudo-replication, all data except for the proportion of broilers were analysed using mixed models containing the fixed effect of the gas treatment (40C90C, 40C60N, and 20C80N) and the cycle as the random effect, using the nlme package [39]. The proportion of broilers that performed a certain behaviour between treatments was compared by means of Pearson’s Chi-squared test. 

## 3. Results

### 3.1. Gas Concentration Assessment

Broiler chickens subjected to 40C90C were exposed to a CO_2_ concentration below 40% by volume in atmospheric air during the first phase, in all cycles (38.1 ± 0.1%). During the second phase, CO_2_ was kept higher than 90% (92.2 ± 0.6%) and residual O_2_ was lower than 2% by volume (1.0 ± 0.1%). On the other hand, broiler chickens subjected to gas mixtures of CO_2_ with inert gases were exposed to CO_2_ concentrations at 36.3 ± 1.1% in 40C60N and 18.0 ± 0.3% in 20C80N, while the O_2_ mean value was below 2% by volume during the 4 min of exposure (40C60N: 1.6 ± 0.3%; 20C80N: 1.9 ± 0.3%). However, the anoxic atmosphere (<2% of O_2_) was steadier over time in 40C60N compared to 20C80N.

### 3.2. Brain Activity Assessment

Brain activity before and during the gas stunning procedure was recorded via EEG in 50 broiler chickens, generating one trace per bird (40C90C: *n =* 16; 40C60N: *n =* 16 and 20C80N: *n =* 18). Twenty-seven out of these 50 EEG records were discarded: the first 10 records from 40C90C due to the low quality of the electrodes used, one of them due to interference from eyelid movement preventing the selection of several 1 s epochs on the baseline; two due to disconnection of the EEG equipment during the exposure to the treatment; and 24 due to recording issues resulting in low-quality records (further explained in discussion section). Hence, EEG analysis was performed on the 23 remaining records (40C90C: *n =* 0; 40C60N: *n =* 14; 20C80N: *n =* 9). 

The time to the onset of LOC and brain death for the 40C60N and 20C80N treatments is summarized in Table 3. The time to the onset of LOC did not differ significantly between 40C60N and 20C80N. However, the time to brain death was similar between the two treatments when the isoelectric pattern was visually identified (*p* > 0.05) but statistically different when spectral analysis was performed (*p* < 0.05).

### 3.3. Behavioural Assessment

#### 3.3.1. Behavioural Assessment of Loss of Posture and Motionlessness

The time to the onset of LOP and motionlessness with respect to the three experimental treatments is summarized in Table 4. Broiler chickens exposed to 40C60N and 20C80N took a similar amount of time to lose posture (21.0 ± 4.5 s, *p* = 0.357) but, significantly, 2.8-fold less time compared to 40C90C (59.2 ± 21.9 s, *p* < 0.001). Likewise, 40C60N and 20C80N took a similar amount of time to remain motionless (68.2 ± 10.3 s, *p* = 0.282) but, significantly, 2.5-fold less time compared to 40C90C (168.8 ± 27.2, *p* < 0.001). It is noteworthy that the range of time to LOP and motionlessness was broader in 40C90C than in 40C60N and 20C80N, while the 40C60N and 20C80N broiler chickens showed less variability. In addition, two broiler chickens exposed to 40C90C lost posture at 144 and 156 s (after 2 min); therefore, they were still conscious when the lift descended to a CO_2_ concentration higher than 40% during the second phase. Furthermore, the latest onset of motionlessness was, in all cases, before the end of the exposure (240 s), indicating that all birds were dead before the end of the process.

#### 3.3.2. Behavioural Assessment before Loss of Posture

A general overview of individual broiler chickens exposed to AIR and gas treatments is shown in Figure 3 and Figure 4, respectively. 

The proportion of birds performing the behaviours, the number of events per bird, and the total duration before LOP, according to the experimental treatment used, are shown in Table 5. 

Sitting, standing, walking, and head shaking were behaviours that broiler chickens performed when exposed to AIR and to all experimental gas treatments before LOP. However, in AIR, all birds sat (100%) and only some stood (30%) at a certain point in time and very few walked (4%), while the proportion of birds that stood and walked increased significantly in all three gas treatments (see Table 5) (*p* < 0.05). In AIR, 2 out of 100 broilers showed head shaking once or twice and the proportion of birds performing head shaking in AIR differed significantly from the experimental gas treatments (2% vs. 100%; *p* < 0.001). No broiler chickens exposed to AIR exhibited ataxia, deep inhalation, gasping, jumping, wing flapping, or high-pitch vocalisations (HPVs). 

When considering only the three gas treatments (i.e., leaving AIR aside), all birds showed head shaking and deep inhalation, while gasping was only performed by some birds in 40C90C and never in 40C60N and 20C80N. When comparing the number of events per behaviour and bird, there was a tendency towards fewer head shakes in 40C60N and 20C80N compared to 40C90C (*p* < 0.10). In addition, deep inhalation events were dramatically reduced in 40C60N and 20C80N compared to 40C90C (*p* < 0.001) but were still present in 40C60N, with a tendency towards fewer events than in 20C80N (*p* = 0.088). The lowest prevalence of vocalizations was found in 40C90C, but the number of HPVs was reduced in 40C60N compared to 40C90C and 20C80N (*p* < 0.001), and was similar between 40C90C and 20C80N (*p* = 0.254). 

Except for wing flapping, the other behaviours assessed occurred in a similar proportion of birds in all gas treatments (*p* > 0.05). However, the 40C60N and 20C80N birds spent significantly less time sitting, standing, and walking (*p* < 0.05) and had fewer events per birds compared to 40C90C (*p* < 0.01). The proportion of broilers showing wing flapping was higher and the duration of wing flapping was longer in 20C80N compared to 40C60N and 40C90C (*p* < 0.01), and in 40C60N compared to the 40C90C treatment (*p* < 0.01). However, it was the broilers exposed to 40C60N that showed the lowest number of events compared to 40C90C and 20C80N. Ataxia in 40C60N was significantly shorter compared to 40C90C (*p* < 0.001), and in 40C60N it tended to be shorter than in 20C80N (*p* = 0.075), while 20C80N also tended to have shorter ataxia than 40C90C (*p* = 0.062).

The order in which each of the behaviours appeared for the first time before LOP is shown in Figure 5. As can be observed, there is a pattern, in which the first behaviours displayed in response to gas treatments are head shaking or deep inhalation. Next, the birds begin to walk, vocalize, become ataxic, and eventually flap their wings. Gasping was only observed in 40C90C, and it was the last behaviour observed before LOP.

#### 3.3.3. Behavioural Assessment after Loss of Posture

The proportion of birds performing different behaviours, the number of events per bird, and their total duration after LOP, according to the experimental treatment used, are shown in Table 6. 

Between LOP and motionlessness, the behaviours observed were gasping, jumping, leg flapping, wing flapping, and HPV. Leg paddling was the most commonly observed behaviour, occurring in almost all birds regardless of gas treatment. Although the proportion of birds showing leg paddling was similar between treatments, 40C90C showed significantly fewer events per bird compared to 20C80N (*p* = 0.041), but a similar number to 40C60N (*p* = 0.203) and, moreover, with a shorter total duration compared to 40C60N and 20C80N (*p* < 0.01).

The lowest proportion of birds performing gasping was found in 40C60N and 20C80N, along with a lower number of gasps per animal, compared to 40C90C (*p* < 0.001). The proportion of birds showing wing flapping was similar between treatments (*p* > 0.05). However, broilers in 40C90C performed fewer wing flapping events compared to 40C60N (*p* < 0.001), and both 40C90C and 40C60N performed fewer events than 20C80N (*p* < 0.001). In addition, 40C90C was the treatment during which broiler chickens performed wing flapping for the least amount of time (*p* < 0.001), while 20C80N had the longest total duration of wing flapping (*p* < 0.001). The proportion of birds that vocalized was similar between treatments and so was the number of vocalizations per bird (*p* > 0.05). Jumping was the least-observed behaviour in all treatments, although the proportion of birds jumping was lower in both 40C90C and 40C60N compared to 20C80N (*p* < 0.01), and 40C90C tended to show fewer events than 40C60N (*p* = 0.099) and significantly fewer than 40C60N (*p* < 0.001).

#### 3.3.4. Relationships between EEG and Behaviour

The time it took for F50 reduction fell within the range of the LOP. Hence, it was considered that the LOP was the beginning of the LOC. In the 40C60N treatment, the F50 values remained below 50% of baseline during the entire gas exposure period. In contrast, the F50 values of the 20C80N treatment exhibited an increase over the 50% baseline after LOP and remained elevated until the end of the gas exposure period. 

During the gas exposure, there was a 65% increase in the proportion of Delta frequencies compared to the baseline. In the EEGs of birds exposed to 20C80N, the proportion of Delta frequencies initially increased but then decreased after 75 s of gas exposure. In the 20C80N treatment, there was an increase in the proportion of Beta frequencies after LOP. The decrease in Delta and increase in Beta frequencies observed in the 20C80N treatment are not biologically relevant, as they corresponded to the times of the isoelectric pattern. The proportion of Gamma frequencies, which are associated with background noise, remained below 2%. The 65% increase in the proportion of Delta frequencies coincided with the time of statistical dispersion until LOP in both the 40C60N and 20C80N treatments. 

As expected, the baseline power spectrum in 40C60N and 20C80N demonstrated a greater dominance of frequencies above 4 Hz, suggesting a state of consciousness. In contrast, the isoelectric power spectrum in the 40C60N treatment exhibited no power, suggesting permanent unconsciousness or brain death. In the case of the 20C80N treatment, the power spectrum during the isoelectric pattern had minimal proportion of 25 Hz. These results explain the increase in both F50 and Beta frequencies observed during the isoelectric pattern of the 20C80N treatment. However, during the isoelectric pattern, the residual power from the EEG is biologically insignificant. Hence, the power spectrum confirmed the non-biological relevance of this increase to the 20C80N treatment. 

## 4. Discussion

### 4.1. Gas Mixture Concentration and Stability Assessment

The concentrations of both CO_2_ and O_2_ stayed stable during the cycles, and within the limits required by the European Union regulation for the protection of animals at the time of killing, across each gas treatment [1].

### 4.2. Brain Activity Assessment

The purpose of stunning is to induce a temporary or permanent disruption of brain function, rendering the animal unconscious and insensitive to pain. In contrast to electrical and mechanical stunning methods, gas stunning does not immediately result in a loss of consciousness. Instead, there is a delay between initiation and the onset of unconsciousness, known as the induction phase. 

EEG is considered the most accurate approach for assessing the transition from consciousness to unconsciousness [22,23,28,30,32,40]. However, its accuracy depends on the quality of the signal. Factors influencing signal quality include the type of electrode, the quality of the recording equipment, the sample rate used by the equipment, signal amplification, noise filtering, the size of the animal’s brain, and the placement of the electrodes (i.e., subcutaneous or intracranial). Hence, there is no established threshold to determine LOC in poultry subjected to CAS. During anaesthesia monitoring, it is feasible to establish a threshold for the unconscious state using an index, such as the Bispectral Index (BIS) [20]. The BIS is computed from spectral variables (Ptot, F50, F95, and frequencies) within an algorithm, generating a numerical value ranging from 0 to 100, where 0 signifies death and 100 indicates consciousness. The threshold for unconsciousness is defined as a BIS below 60 [20,41]. However, the practical application of the BIS is not feasible for small farmed animals in the context of slaughter due to the absence of an amplifier of the signal, noise filtering, the small size of their brain, and contamination of signal due to muscular activity [20,41].

Additionally, the movements of conscious animals can be restricted but not eliminated. In fact, even blinking can lead to the exclusion of records. Another factor influencing the quality of the outcomes is the type of analysis conducted, which relies on the evaluator’s expertise, particularly in subjective analyses like the visual examination of filtered records. The reasons for losing all data from the EEG in the 40C90C treatment are still unknown, but perhaps the quality of the electrodes used on the first day of the experiment could have been the cause. In accordance with authors’ experience, a change in a batch of commercial electrodes can have a negative impact on the signal quality, due to the quality of the metals used in the needle, the welding, or the wire. The use of costumed electrodes can be an alternative as an attempt to increase the signal quality. The use of costumed electrodes had been used for EEG in animals with the smallest of brains, like fish, with satisfactory results [42]. 

The time elapsed until the onset of LOC in 40C60N and 20C80N was determined using EEG spectral variables. Specifically, the onset of LOC was identified as when the F50 decreased below 50% and the Delta frequencies increased to above 65% of the baseline. It is important to highlight that the chosen threshold values in this study differ from those used in previous research studies. For example, in broiler chickens exposed to LAPS, LOC was determined when the F50 dropped below 7 Hz, compared to baseline values of 20 Hz [24], or when the F50 was reduced to below 75% of the baseline [27]. The different thresholds used in our study compared to LAPS are attributed to the type of electrodes used. In our study, subcutaneous electrodes were used, while other authors [24] used implanted electrodes placed through the skull, resulting in outputs on a different scale. Although subcutaneous electrodes provide lower-quality EEG recordings compared to implanted electrodes, they are less invasive for the animals as they do not require surgery or recovery time prior to the experiment. In the present study, despite the use of subcutaneous electrodes, the contribution of Gamma frequencies to the Ptot remained at an average of 0.1%, indicating irrelevant interference from background noise [27]. This confirms the quality of the EEG recordings and the reliability of the obtained results in determining the time elapsed until the onset of LOC and death.

General anaesthesia serves as an ideal model for understanding the EEG waveform alterations observed during the transition from a conscious to an unconscious state [23]. During general anaesthesia stages, there is a predominance of the Delta and Theta frequency bands [22]. F50 is particularly sensitive to changes in lower frequencies, while F95 is more responsive to shifts towards higher frequencies [20]. Reductions in the Ptot and F50 are well-established indicators correlated with clinical signs of a loss of consciousness and anaesthesia in animals [43,44].

In the study conducted by Sandercock et al. [22], a reduction in F50 was observed during inhalational anaesthesia, using a face mask with a sevoflurane vaporizer at a concentration of 8%, in hens and turkeys. So far, there are no studies that have reported a specific threshold for Delta frequency proportions. Previous reports have described the analysis of the Delta frequency on EEG as a visual change in the EEG trace, without an objective threshold [19,33]. Therefore, a decrease below 50% of the F50 and an increase above 65% of the Delta frequency proportion can be considered potential indicators of LOC during EEG recording in broiler chickens exposed to gas stunning.

The time elapsed until the onset of LOC found in 40C60N and 20C80N was similar to those reported in previous studies [16,17]. McKeegan et al. [16] reported that the onset of LOC in broiler chickens exposed to a gas mixture of 40% CO_2_ and 60% N_2_ was at 23 ± 4 s. In our study, it occurred, interestingly, at a similar timing, at 25.7 ± 7 s. This slight difference may be due to the use of a pit pre-filled with gas instead of the closed cabinet (not pre-filled but flushed) used in McKeegan [16] and, therefore, the reduction in available O_2_ was reported to be not instantaneous; thus, there was a slight delay in reaching the desired modified atmosphere. Coenen et al. [17] reported a LOC in broiler chickens exposed to a gas mixture of 30% CO_2_ and 70% N_2_ in a tunnel system at 34 ± 12 s (determined by visual analysis of EEG waveforms when the trace was isoelectric). 

The visual interpretation of EEGs can be subjective, and various studies have employed different waveform patterns to determine the LOC, such as suppressed, isoelectric, high-amplitude low-frequency (HALF), and transitional states [16,19,45]. For this reason, we only use visual analysis as a proxy for death, relying on the consolidated indicator of the isoelectric pattern [19,24,27,28,30,33,34]. In addition to visual analysis, spectral analysis provides a quantitative approach for assessing the LOC by generating numerical results. The utilization of numerical variables derived from EEGs has gained attention in similar experimental studies in recent years [23,27,35,40]. This complementary approach enhances the accuracy and reliability of assessing LOC. 

Permanent unconsciousness or brain death was assessed using EEG through visual analysis and spectral analysis. The average time for reaching irreversible loss of brain function and brain stem death, and observing the isoelectric pattern visually, was estimated to be 69.8 ± 11.9 s in the 40C60N treatment and 66.3 ± 8.1 s in the 20C80N treatment. The high variability observed in 20C80N for death via the spectral analysis may be attributed to signal degradation. Signal degradation refers to slight changes in frequencies when there is a low Ptot. These changes can influence the frequency contribution and the Ptot itself. Hence, determining the point of death may take longer. However, since the Ptot is low, these changes do not represent any biological significance and rather indicate signal degradation. In such cases, visualizing the isoelectric pattern could provide a more accurate assessment. Both visual and spectral analysis methods demonstrated the irreversible cessation of brain activity and brain stem function [19,22,23,27]. In similar studies on broiler chickens exposed to gas stunning in an experimental chamber using 40% CO_2_ and 60% N_2_ via flushing, the time to death estimated by a visual analysis of the EEG was 67.8 ± 4.6 s [16].

Raj et al. [46] reported a slightly different onset time of isoelectric EEG at 58 ± 2.3 s in broiler chickens exposed to a gas mixture (30% CO_2_ + 60% Argon + 10% air) in a pre-filled box, which may be attributed to the time it takes for the lift to descend into the pit (23 s) and reach the target gas concentration. In contrast, McKeegan et al. [16] visually observed the onset of death at 80.7 s through the isoelectric pattern on the EEG and the absence of motion in broiler chickens exposed to 40% CO_2_ and 60% N_2_ in a gas-flushed closed cabinet. If the exposure time to the gas is insufficient to induce brain death and the birds can breathe atmospheric air, they may quickly regain consciousness [12]. Our results suggested that a four-minute exposure to the 40C60N and 20C80N gas mixtures induced permanent unconsciousness in all birds, eliminating the possibility of them regaining consciousness. The integration of the visual and spectral analysis of EEG proved to be a reliable method for accurately estimating brain death in broiler chickens exposed to gas stunning.

The present study intended to record EEG traces from 50 broiler chickens. However, 28 out of 50 (56%) EEG records were unsuitable for analysis due to the loss of electrodes during gas exposure or non-readable EEG activity. This decrease in the sample size is common when EEG is performed. Previous studies have reported a loss of readable EEGs in 9–71% of animals [33,42,47].

### 4.3. Behavioural Assessment

#### 4.3.1. Behavioural Assessment of Loss of Posture and Motionlessness

Broiler chickens differed in the elapsed time to LOP and motionlessness according to the experimental gas treatment they were subjected to. In particular, exposure to 40C60N or 20C80N not only drastically reduced the time until the birds lost posture and became motionless, but there was also much less inter-individual variability in the time to LOP compared to 40C90C. The high inter-individual variability in the time to LOP in 40C90C represents a serious welfare risk if the birds still remain conscious when the first phase has finished, and certain birds are therefore exposed to more than 40% CO_2_ during the second phase while conscious, as was the case for 2 out of 76 broiler chickens in the present study. It is known that the inhalation of concentrations of above 40% CO_2_ in conscious chickens generates a very painful mucosal stimulus [15,48].

#### 4.3.2. Behavioural Assessment before Loss of Posture

Since LOP is considered the behavioural indicator of the onset of LOC [10,26,31,49], all the behaviours observed before the LOP occur in birds that are conscious and therefore may be potential indicators of pain, distress, or dyspnoea caused by the inhalation of the gas or gas mixture. In order to discern whether the descent into the pit per se caused aversive behaviours in broiler chickens and caused confusion in the results, first, the behaviour of the birds in the gas stunning equipment was assessed while they were breathing atmospheric air (AIR), and then again when subjected to experimental treatments. In AIR, most chickens remained sitting, while some were standing, few walked, and very few exhibited head shaking once or twice while descending into the pit. Although these behaviours were also observed in the other three experimental treatments, the proportion of birds performing sitting and walking was higher in the gas mixtures compared to AIR, which may indicate a behavioural change pattern that might be related to fear. The cause of head shaking observed in only two birds in the AIR group remains unclear. Unlike the head shaking observed in the gas mixtures, the head shaking in AIR was not associated with a sound of sneezing. It is possible that these birds are sensitive to new stimuli [11,50], like the descent of the lift, or that they are attempting to self-activate after a period of rest [48]. The almost minimal proportion of birds exhibiting head shaking in the AIR treatment, along with the absence of other considered aversive behaviours (i.e., deep inhalation, gasping, wing flapping, HPVs, ataxia) suggests that descending into the pit did not induce aversion in broiler chickens, unlike the behaviour that was observed during the three experimental treatments. 

Head shaking and deep inhalation was performed at least once by all birds from all gas treatments, and head shaking was associated with a sneezing sound, but gasping was only observed in some birds exposed to 40C90C. Head shaking, deep inhalation, and gasping are associated with mucosal irritation, dyspnoea, and breathlessness (“air hunger”) and, thus, a reduction in welfare during gas stunning [27]. Head shaking and deep inhalation were the first aversive behaviours displayed during the stunning process. While head shaking is associated with an unpleasant stimulus caused by the activation of chemoreceptors sensitive to CO_2_ in respiratory tract, deep inhalation and gasping are associated with hyperventilation during CO_2_ stunning; however, these responses have also been observed with the inhalation of inert gases alone, such as argon [49], and are related to respiratory distress [26]. So far, no scientific study on gas stunning in broilers has included vocalisations as a behaviour to be assessed in the ethogram [15,16,30,51]. However, the vocalisations recorded in this study are particularly high-pitched and suggestive of fear and/or pain, if heard before LOP. 

Comparing the three experimental gas treatments, both 40C60N and 20C80N showed fewer transitions between locomotor behaviours (sitting, standing, and walking) and these were of a shorter duration than 40C90C, because the time to LOP in these treatments was 2.8-fold shorter (40C60N and 20C80N: 21.0 ± 4.5 s vs. 40C90C: 59.2 ± 21.9 s), so the likelihood of repeating these locomotor behaviours is lower. Despite this shorter time to LOP, both 20C80N and 40C60N had a similar proportion of birds displaying head shaking, deep inhalation, and HPVs. In addition, 20C80N had similar number of head shakes and HPVs but fewer deep inhalations and no gasping compared to 40C90C. In contrast, 40C60N showed a tendency to have less head shaking, less HPVs, and no gasping compared to 40C90C. Therefore, both 20C80N and 40C60N seems to cause less aversion than 40C90C.

The proportion of chickens that flapped their wings before LOP varied depending on the experimental treatment. Nitrogen-containing gas mixtures caused a higher proportion of birds to flap their wings before losing posture than 40C90C. The higher the nitrogen concentration in the gas mixture, the higher the proportion of birds showing wing flapping. This could be due to the fact that anoxic environments lead to an increase in wing flapping, although CO_2_ has an anaesthetic effect on birds and, therefore, when a higher CO_2_ concentration and anoxic environment are combined (as in 40C60N compared to 20C80N) it can result in a calmer induction of unconsciousness. The occurrence of wing flapping in the gas stunning equipment is a welfare concern, since it may cause injuries and pain to the other birds that have not yet lost consciousness [5,16].

In addition, in 40C60N or 20C80N there was no risk that birds inhaled CO_2_ concentrations above 40%, as can happen in birds that are still conscious when the 2nd phase of 40C90C starts. Therefore, in 40C60N or 20C80N the experience of severe pain in the mucosa is mitigated. 

Taking all this into consideration, it seems that the fastest induction to unconsciousness was achieved with 40C60N, which was also the least aversive gas mixture. Moreover, the risk of inhaling a CO_2_ concentration above 40% (known to cause severe pain in conscious birds due to the activation of chemoreceptors in mucous membranes), as can occur in 40C90C, is mitigated. On the other hand, 20C80N also offers a rapid induction of unconsciousness (of a similar time to 40C60N); it appears to be less aversive than 40C90C but slightly more so than 40C60N, and there is also no risk of inhaling a CO_2_ concentration above 40%. Therefore, from an animal welfare point of view, 40C60N appears to be the least aversive of the three treatments tested as our experimental conditions.

#### 4.3.3. Behavioural Assessment after Loss of Posture

Since LOP was the behavioural indicator of the onset of unconsciousness, behaviours after LOP do not represent a welfare concern (e.g., pain, distress, breathlessness) as the bird is presumed to be unconscious. The behaviours observed may be related to convulsions or other involuntary movements rather than aversive behaviours [27,33]. 

The convulsions in broiler chickens exposed to the three experimental treatments were expressed as wing flapping, leg paddling, and jumping due to uncontrolled muscle jerks. Leg paddling and jumping were behaviours observed only after LOP, but leg paddling was observed in almost all birds at this stage. Therefore, leg paddling seems to be the most reliable indicator of convulsions and unconsciousness.

Gasping and HPVs were performed both before and after LOP in gas treatments, indicating that consciousness is not required for their performance. In relation to HPVs, birds do not need to contract the vocal cords in their larynx to produce sound, unlike terrestrial mammals. Additionally, the eight air sacs present in birds are expansions of the respiratory tree. During convulsions, the muscular jerks can lead to the passing of air from the air sac to the lung and then through the syrinx, causing vocalizations in unconscious birds. HPVs after LOP are presumed to be a consequence of air movement through the syrinx caused by convulsions (muscle jerks). One might think that HPVs could also suggest pain or aversion after LOP. However, during the experiment, even once the dead chickens were removed from the pit and piled up, we heard squeaks due to the compression exerted by the weight of one carcass on top of another.

The number of HPVs heard after LOP was higher in the 20C80N treatment, which is consistent with the higher number of wing flapping events observed in this treatment. Similarly, as they occurred before LOP, anoxic environments also lead to an increase in wing flapping and leg paddling after LOP, but the higher the CO_2_ concentration, the lower the convulsions expressed as wing flapping due to the anaesthetic effect of CO_2_ on birds. The occurrence of wing flapping in the gas stunning equipment may be a welfare concern since it may cause injuries and pain to other birds that have not yet lost consciousness [5,16]. This finding is consistent with a study by Gent et al. [10], which reported a longer duration of wing flapping after the LOP in broiler chickens exposed to N_2_ (19.7 s, on average) compared to CO_2_ (7.1 s, on average). Further research could investigate the effect of different gas mixtures on the meat quality of broiler chickens subjected to stunning. 

### 4.4. Relationship between Brain Activity and Behavioural Assessment

The relationship between LOC and death events in the EEG activity and behavioural observations in unrestrained animals is crucial for interpreting behavioural indicators. Nonetheless, there was no statistical correlation between LOC and LOP in this study because brain activity and behaviour were not assessed in the same animal (r = −0.091; *p* = 0.779). Benson et al. [31] surgically implanted wireless EEG electrodes into broiler chickens and, once recovered, the chickens were unrestricted and exposed to isoflurane anaesthesia to correlate their LOC determined by brain activity with their LOP. However, this correlation was not statistically significant (r = 0.150; *p* > 0.05). The authors concluded that the assessment of LOP has a certain inaccuracy as it depends on the subjectivity of the observer, who defines when the bird ceases to maintain a sitting position or neck tension. In addition, the determination of LOP was sometimes hindered in cases where birds were huddled against the walls of the chamber, artificially providing them with support. However, they concluded that LOP can be utilized as a proxy for the onset of LOC.

In the present study, the range of time to LOC and brain death (i.e., an isoelectric pattern during EEG) was similar to the range of time to LOP and motionlessness in 40C60N and 20C80N, respectively. Therefore, the results suggest that LOP and motionlessness can be used as behavioural indicators to estimate LOC and brain death, respectively, when EEG recording is not a possibility (e.g., in commercial slaughterhouses or depopulation conditions). The comparison between LOP and LOC was carried out only in the 40C60N and 20C80N treatments. We considered 40C60N the less aversive treatment, due to the smaller proportion of birds performing vocalizations before LOP and the faster LOC. 

The use of video cameras to monitor the behaviour of birds during stunning procedures under commercial conditions is of the utmost importance, as well as operators being trained to detect indicators of loss of consciousness, death, aversion, and convulsion. However, in current CAS designs, even if video cameras are present, it is not always possible to monitor the behaviour of the animals, as the birds are often stunned directly in their transport containers, where the observer’s visibility and the mobility of the birds is greatly reduced. In current designs where birds are removed from their transport containers before being introduced into the gas stunning systems, the monitoring of the behaviour of the birds is also hindered as they are usually stunned in tunnel-type systems where the birds are conveyed from one end of the system to the other while exposed to gas concentrations, so the images observed via video camera of a specific individual are of a very limited amount of time. 

## 5. Conclusions

The exposure of broiler chickens to 40C90C, 40C60N, or 20C80N does not induce immediate unconsciousness. Regardless of the gas mixture tested, all broiler chickens experienced aversion during the induction of a loss of consciousness. The exposure to 40C60N and 20C80N not only decreased dramatically the time to the induction of LOC and death, but also did so with less variability in the elapsed time between individuals compared to 40C90C. The 40C90C birds not only experienced more aversion during the induction of LOC but were also at risk of remaining conscious when the CO_2_ concentration was increased in the 2nd phase. From an animal welfare point of view, 40C60N was the least aversive of the three treatments tested, followed by 20C80N and 40C90C. Further research is required to explore alternative gases or gas mixtures that can minimize or eliminate aversive responses during the induction of unconsciousness to improve broiler chicken welfare during stunning. 

## Figures and Tables

**Figure 1 animals-14-00486-f001:**
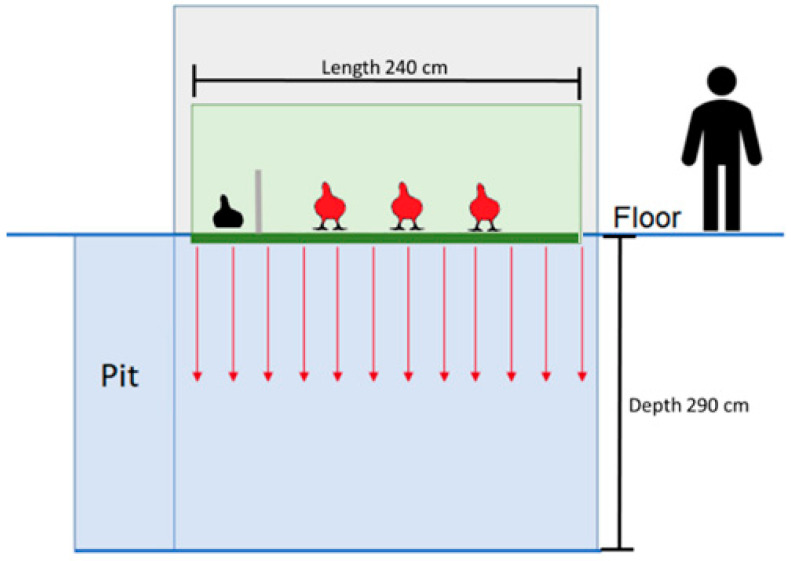
Representation of the distribution of broiler chickens in the pit-type gas stunning system per stunning cycle (dips into the pit). In each cycle, four birds were stunned, one bird had its brain activity recorded via electroencephalography (bird highlighted in black) and three birds were recorded with video cameras to assess their behaviour (birds highlighted in red). The bird whose brain activity was assessed was separated from the rest of the chickens by a transparent methacrylate wall.

**Figure 2 animals-14-00486-f002:**
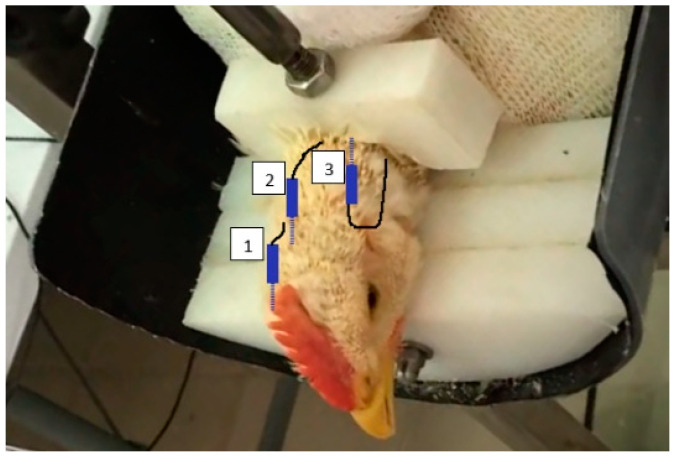
Representation of needle electrodes’ positions for brain activity assessment via electroencephalography in feather-shaved broiler chickens with an (1) active electrode, (2) reference electrode, and (3) ground electrode.

**Figure 3 animals-14-00486-f003:**
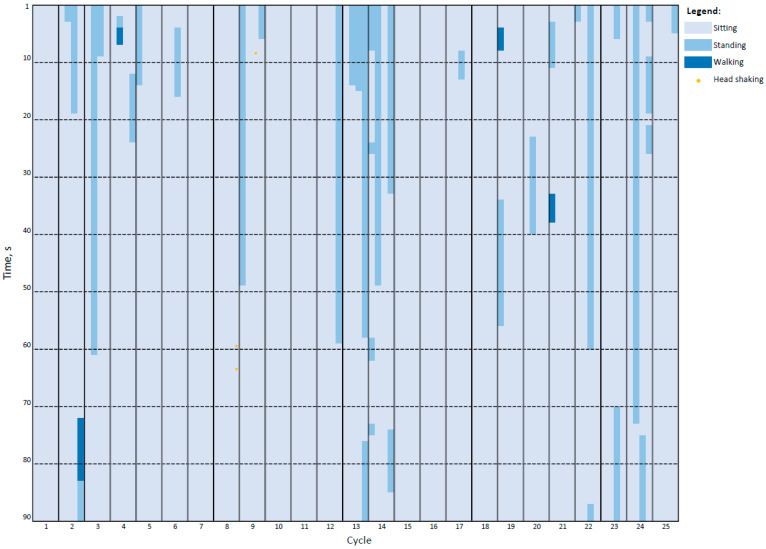
Behavioural plot of broiler chickens (*n =* 100) exposed to atmospheric air in a pit-type gas stunning system. The graphical plot shows the behaviour of the birds in 1 s bins. Segments of 10s appear as horizontal dashed lines, whereas cycles (dips into the pit) are displayed as vertical lines. Four birds were used per cycle and each bird’s behaviour is represented by coloured vertical segments based on the colour coding shown in the legend.

**Figure 4 animals-14-00486-f004:**
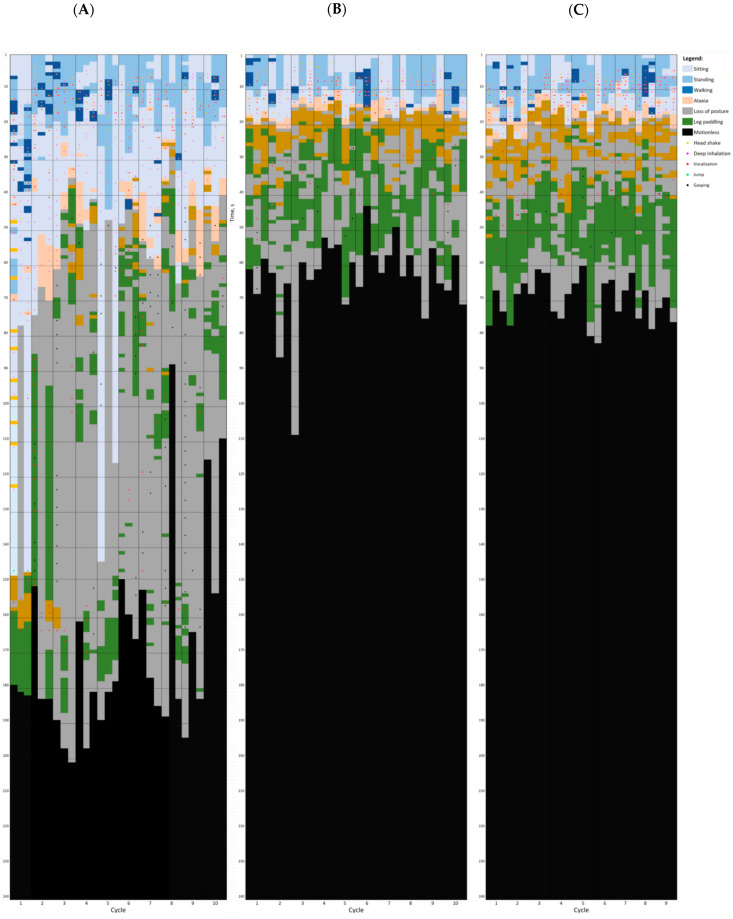
Behavioural plot of a sample of broiler chickens exposed to different gas stunning treatments: (**A**) CO_2_ in two phases, the 1st phase with <40% CO_2_ by volume in air for 2 min and 2nd phase with >90% CO_2_ for 2 min (*n =* 30); (**B**) gas mixture of 40% CO_2_ and 60% nitrogen (N_2_) with less than 2% O_2_ for 4 min (*n =* 30); and (**C**) gas mixture of 20% CO_2_ and 80% N_2_ with less than 2% O_2_ for 4 min in a pit-type gas stunning system (*n =* 27). The graphical plot shows the behaviour of the birds in 1 s bins. Segments of 10s appear as horizontal dashed lines whereas cycles (dips into the pit) are displayed as vertical lines. Three birds were used per cycle and each bird’s behaviour is represented by coloured vertical segments based on the colour coding shown in the legend.

**Figure 5 animals-14-00486-f005:**
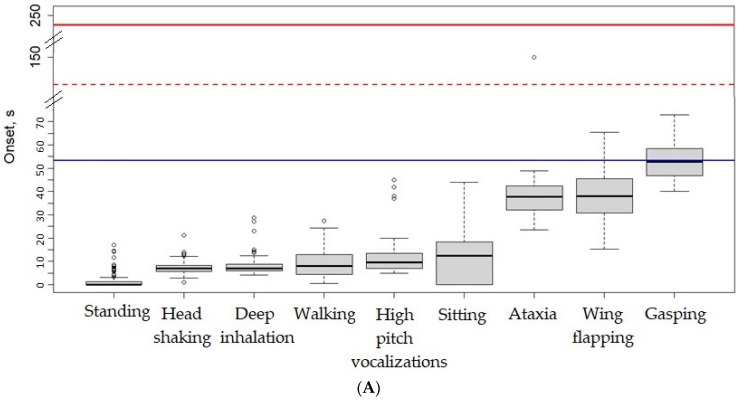
Boxplot of time to the onset of the different behaviours in broiler chickens before loss of posture, in sequence, after the exposure to different gas mixtures. Blue line represents median time to loss of posture, dotted red line means the second phase of 40C90C treatment, and red line indicates end of the period of exposure. (**A**) 40C90C: CO_2_ in two phases, the first phase with <40% CO_2_ by volume in air for 2 min and the second phase with >90% CO_2_ for 2 min (*n =* 76, stunned in groups of three except for two times that birds were stunned in groups of two). (**B**) 40C60N: gas mixture of 40% CO_2_ and 60% N_2_ with less than 2% O_2_ for 4 min (*n =* 63, stunned in threes). (**C**) 20C80N: gas mixture of 20% CO_2_ and 80% N_2_ with less than 2% O_2_ for 4 min (*n =* 54, stunned in threes).

**Table 1 animals-14-00486-t001:** Schedule of the gas stunning treatments applied along the 5 d experimental period to broiler chickens.

Item	Day 1	Day 2		Day 3		Day 4		Day 5
Session	1500 to 1900 h	800 to 1200 h	1500 to 1900 h	800 to 1200 h	1500 to 1900 h	800 to 1200 h	1500 to 1900 h	800 to 1200 h
Treatment	AIR	40C90C	40C60N	40C60N	20C80N	20C80N	40C90C	40C90C
cycles, *n*	25	10	10	11	9	9	8	8
Total birds, *n*	100	30	35	44	36	36	32	30
Birds/cycle	4	3	3–4 *	4	4	4	4	3–4 **
Brain activity assessment, *n*	0	0	5	11	9	9	8	8
Behavioural assessment, *n*	100	30	30	33	27	27	24	22
Cycles, *n*	25	10	10	11	9	9	8	8

* Five cycles with three birds used in each (none for EEG assessment) and five cycles with four birds used in each (one for EEG assessment and three for behavioural assessment). ** Six cycles with four birds used in each (one for EEG assessment and three for behavioural assessment) and two cycles with three birds used in each (one for EEG assessment and two for behavioural assessment).

**Table 2 animals-14-00486-t002:** Ethogram used to assess the behaviour of broiler chickens subjected to different experimental gas stunning treatments.

Behaviour	Description	Adapted from
Loss of posture	Cessation of standing, with the head resting against either the floor or wall of the gas stunning system.	[10,26]
Motionlessness	Limp carcass, with the bird being completely still including the cessation of visible breathing movements.	[27]
Sitting	Legs underneath the body and wings relaxed against body.	[28]
Standing	On their feet with the body fully or partly lifted off of the ground.	[24]
Walking	Moving forward at a regular pace.	[29]
Ataxia	Uncoordinated walking with exaggerated lateral movement or fluttering when standing to maintain posture.	[10]
Deep inhalation	Wide open-mouth breathing with neck extension.	[10]
Gasping	Opening and closing mouth without neck extension and with reduced frequency compared to physiological breathing.	[10]
Head shaking	Rapid side-to-side movement of the head, which occurs whilst the bird is standing, walking, or sitting.	[10]
Jumping	Any vertical movement from a plantar stance, resulting in both feet leaving contact with the floor.	[10]
Leg paddling	Leg movements in the air or towards the ground depending on the body position of the bird. It can also be determined by an alternating upwards and downwards movement of the body if bird is lying sternal.	[28]
Wing flapping	Bouts of fast, short flapping, rapid movement of the wings.	[10,30]
High-pitch vocalisations	Single or repeated short and loud shrieking (screaming).	[4]

**Table 3 animals-14-00486-t003:** Time to onset of loss of consciousness and death, determined by electroencephalography of broiler chickens submitted to different experimental gas stunning treatments.

		Treatments			
Analysis	Outcome	40C90C	40C60N	20C80N	*p*-Value
Spectral	Loss of consciousness, s *	NA	25.7 ± 7.0	20.7 ± 6.6	0.144
Spectral	Death, s **	NA	65.8 ± 14.1 ^b^	122 ± 53.2 ^a^	0.048
Visual	Death, s ***	NA	69.8 ± 11.9	66.3 ± 8.1	0.456

40C90C: CO_2_ in two phases entailed a first phase with <40% CO_2_ by volume in air for 2 min and a second phase with >90% CO_2_ for 2 min (*n =* 0); 40C60N: a gas mixture of 40% CO_2_ and 60% nitrogen (N_2_) with less than 2% O_2_ for 4 min (*n =* 14); and 20C80N: a gas mixture of 20% CO_2_ and 80% N_2_ with less than 2% O_2_ for 4 min (*n =* 9). *, based on spectral analysis (reduction of 50% of F50 or increase of 65% of Delta contribution). **, based on spectral analysis (reduction of 90% Ptot). ***, not based on spectral analysis but visual (isoelectric pattern). NA means not available. Different superscripts in the same row indicate a significant statistical difference between treatments (*p* < 0.05).

**Table 4 animals-14-00486-t004:** Time to onset of loss of posture and motionlessness of broiler chickens, expressed as mean (min–max), when submitted to different experimental gas stunning treatments.

	Treatment				
Behaviour	40C90C	40C60N	20C80N	SE	*p*-Value
Loss of posture	59.2 (26.0–156.5] ^a^	19.8 (14.0–30.8] ^b^	22.3 (15.8–37.0] ^b^	2.7	<0.001
Motionless	168.8 (89.0–212.7] ^a^	66.1 (43.0–108.0] ^b^	70.4 (45.2–88.5] ^b^	3.5	<0.001

40C90C: CO_2_ in two phases entailed a first phase with <40% CO_2_ by volume in air for 2 min and a second phase with >90% CO_2_ for 2 min (*n =* 76, stunned in groups of three except for two times that birds were stunned in groups of two); 40C60N: a gas mixture of 40% CO_2_ and 60% N_2_ with less than 2% O_2_ for 4 min (*n =* 63, stunned in groups of three); and 20C80N: a gas mixture of 20% CO_2_ and 80% N_2_ with less than 2% O_2_ for 4 min (*n =* 54, stunned in groups of three). Different superscripts in the same row indicate a significant statistical difference between treatments (*p* < 0.05).

**Table 5 animals-14-00486-t005:** Proportion of broiler chickens that showed different behaviours and the number of events per individual, expressed as mean (min–max), when inhaling the experimental gas stunning treatments. Birds were tested before losing posture.

		Treatment				
Item	Behaviour	40C90C	40C60N	20C80N	SE	*p*-Value
Proportion, (%)	Head shaking	76/76 (100%)	63/63 (100%)	54/54 (100%)	-	1.000
	Deep inhalation	76/76 (100%)	63/63 (100%)	54/54 (100%)	-	1.000
	HPV	42/76 (55.3%)	43/63 (68.3%)	53/54 (98.1%)	-	0.101
	Gasping	20/76 (26.3%)	0/63 (0.0%)	0/54 (0.0%)	-	-
	Sitting	75/76 (98.7%)	58/63 (92.1%)	50/54 (92.6%)	-	0.951
	Standing	71/76 (93.4%)	51/63 (81.0%)	48/54 (88.9%)	-	0.848
	Walking	45/76 (59.2%)	25/63 (39.7%)	28/54 (51.9%)	-	0.413
	Ataxia	63/76 (82.9%)	55/63 (87.3%)	48/54 (88.8%)	-	0.297
	Wing flapping	20/76 (26.3%) ^c^	35/63 (55.5%) ^b^	43/54 (79.6%) ^a^	-	0.001
Events/bird, *n*	Head shaking	5.4 (1–12)	4.8 (1–9)	5.1 (1–9)	0.3	0.069
	Deep inhalation	9.4 (2–18) ^a^	3.9 (1–9) ^b^	4.8 (0–9) ^b^	0.4	<0.001
	HPV	2.9 (0–12) ^a^	2.0 (0–6) ^b^	3.3 (0–11) ^a^	0.4	0.001
	Gasping	0.7 (0–10)	0 (0–0)	0 (0–0)	0.2	-
	Sitting	2.1 (1–6) ^a^	1.3 (0–3) ^b^	1.6 (0–4) ^ab^	0.2	<0.001
	Standing	2.0 (0–7) ^a^	1.1 (0–3) ^b^	1.3 (0–4) ^b^	0.2	0.004
	Walking	1.2 (0–7) ^a^	0.6 (0–3) ^b^	0.7 (0–3) ^b^	0.2	<0.001
	Wing flapping	0.4 (0–4) ^c^	0.6 (0–2) ^b^	1.0 (0–3) ^a^	0.1	<0.001
Total duration, s	Sitting	35.9 (3.4–135.1) ^a^	7.4 (0–19.5) ^b^	6.9 (0–22.8) ^b^	2.7	<0.001
	Standing	13.4 (0.0–37.9) ^a^	7.5 (0.0–22.2) ^b^	8.5 (0–18.2) ^b^	1.1	<0.001
	Walking	2.6 (0.0–11.8) ^a^	1.5 (0.0–10.5) ^b^	1.5 (0.0–9.3) ^b^	0.5	0.024
	Ataxia	4.6 (0.0–19.7) ^a^	2.7 (0.0–7.1) ^b^	3.6 (0.0–8.2) ^ab^	0.5	<0.001
	Wing flapping	0.7 (0.0–6.8) ^c^	1.1 (0.0–6.2) ^b^	2.3 (0.0–6.5) ^a^	0.3	0.021

40C90C: CO_2_ in two phases, the first phase with <40% CO_2_ by volume in air for 2 min and the second phase with >90% CO_2_ for 2 min (*n =* 76, stunned in threes except for two times that birds were stunned in groups of two); 40C60N: gas mixture of 40% CO_2_ and 60% N_2_ with less than 2% O_2_ for 4 min (*n =* 63, stunned in threes); and 20C80N: gas mixture of 20% CO_2_ and 80% N_2_ with less than 2% O_2_ for 4 min (*n =* 54, stunned in threes). HPV means high-pitch vocalisation. Different superscripts in the same line indicate significant statistical differences between treatments (*p* < 0.05).

**Table 6 animals-14-00486-t006:** Proportion, number of events/bird, and total duration of behaviours observed in broiler chickens exposed to different experimental gas stunning treatments, expressed as mean [min–max], tested after losing posture.

		Treatment				
Item	Behaviour	40C90C	40C60N	20C80N	SE	*p*-Value
Proportion, *n* (%)	Gasping	43/76 (56.6%) ^a^	14/63 (22.2%) ^b^	14/54 (25.9%) ^b^	-	0.009
	Jumping	60/76 (7.9%) ^b^	7/63 (10.5%) ^b^	14/54 (25.0%) ^a^	-	0.003
	Leg paddling	75/76 (98.7%)	63/63 (100%)	54/54 (100%)	-	0.998
	Wing flapping	53/76 (69.7%)	61/63 (96.8%)	53/54 (98.1%)	-	0.320
	HPV	65/76 (85.5%)	53/63 (84.1%)	53/54 (98.1%)	-	0.818
Events/bird, *n*	Gasping	2.5 (0–16) ^a^	0.4 (1–3) ^b^	0.3 (0–3) ^b^	0.4	<0.001
	Jumping	0.1 (0–2) ^b^	0.2 (0–3) ^ab^	0.4 (0–3) ^a^	0.1	<0.001
	Leg paddling	3.9 (0–10) ^b^	4.3 (2–8) ^ab^	4.6 (0–9) ^a^	0.3	0.041
	Wing flapping	1.5 (0–0) ^c^	2.4 (0–7) ^b^	3.1 (0–7) ^a^	0.3	<0.001
	HPV	1.8 (0–10)	1.5 (0–7)	1.8 (0–4)	0.3	0.297
Total duration, s	Leg paddling	14.3 (1.5–36.5) ^b^	18.2 (4.0–39.3) ^a^	19.0 (2.0–36.9) ^a^	1.3	<0.001
	Wing flapping	0.7 (0–6.8) ^c^	1.1 (0–6.2) ^b^	2.3 (0–6.5) ^a^	0.3	<0.001

40C90C: CO_2_ in two phases, the first phase with <40% CO_2_ by volume in air for 2 min and the second phase with >90% CO_2_ for 2 min (*n =* 76, stunned in threes except for two times that birds were stunned in groups of two); 40C60N: a gas mixture of 40% CO_2_ and 60% N_2_ with less than 2% O_2_ for 4 min (*n =* 63, stunned in threes); and 20C80N: a gas mixture of 20% CO_2_ and 80% N_2_ with less than 2% O_2_ for 4 min (*n =* 54, stunned in threes). HPV means high pitch vocalisation. Different superscripts in the same line indicate significant statistical differences between treatments (*p* < 0.05).

## Data Availability

Data sharing upon request.

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
