# Peer review of "Alternatives to Carbon Dioxide in Two Phases for the Improvement of Broiler Chickens’ Welfare during Stunning"

_animals, 2024, doi:10.3390/ani14030486_

Round 1

Reviewer 1 Report

Comments and Suggestions for Authors

Overall, a very interesting, well executed piece of research addressing an important and relevant topic, aiming to improve bird welfare at slaughter. The manuscript is mostly well written, with some minor areas requiring further clarification which have been further detailed in the comments below.

Line 14 and 43. I would suggest the use of the term unconscious rather than unconsciousness.

Line 48-50 – This sentence is missing a word, currently does not make sense.

In the methods section the number of birds allocated to each experimental group is not clear and sometimes inconsistent. It is stated on line 133 that birds are allocated to each group in equal numbers to each treatment, however the n numbers included in lines 134 to 138 are not equal. These n numbers are also different to those included in Table 1. This needs to be clarified.

Line 185 does not make sense.

Tables 4/5/6/7 – all contain subscript letters however there is no clear reference in the legend as to what there are referring to. It is also not clear why the n numbers are different from Table 1.

Line 353 and also 675 – all abbreviations need written out on their first inclusion.

Subheading 3.3.3 is used twice once line 399 and once line 436.

Line 475 – This should be made clear that it is drawn from the behaviour results not from brain activity.

Line 587 – incorrect tense of verb.

Line 591 – Confusing sentence as it referes to increased sitting then also states showing increased locomotor activity. Please reword.

Line 615-617 – Please correct grammar here.

Line 618 – Do the authors mean no previous study here?

Line 706 – Should be ‘than’ rather than ‘that’

Comments on the Quality of English Language

In general the English is acceptable. There are some areas where the grammar/tense of verbs is not correct. I have highlighted some of these in the specific comments, however the article should be carefully reviewed to ensure a high quality of English throughout. 

Author Response

Dear Reviewer,

Find enclosed a word document with our response to your comments.

Kind regards

Reviewer 2 Report

Comments and Suggestions for Authors

This study assessed the animal welfare impact three stunning methods for broilers using EEG and behavioral indicators for welfare. The study addresses an important animal welfare issue and the research design is appropriate. Some study limitations need to be addressed more, and authors need to acknowledge the impact of these limitations. Please find attached the detailed comments. 

Comments on the Quality of English Language

Author Response

Dear Reviewer, we are truly grateful for the high-quality feedback provided. It helps us to improve the manuscript a lot. Please see the atachment.
PS: English was revised by native speaker

Reviewer 3 Report

Comments and Suggestions for Authors

The research, titled " Alternatives to the use of carbon dioxide in two phases for the improvement of broiler chickens’ welfare during stunning" aims to assess the impact of alternative gas mixtures on the welfare of broiler chickens during the slaughter process. This research topic holds significant relevance and intrigue, particularly in light of the growing consumer concern for the welfare of farm animals. The study aligns well with the journal's scope.

This study investigates the use of gas mixtures of carbon dioxide (CO2) associated with nitrogen (N2) as alternatives to the traditional CO2 stunning method in broiler chickens during slaughter. The aim is to improve the welfare of the chickens by assessing the time to unconsciousness and death, as well as characterizing aversive responses to the treatments. The results show that the gas mixtures 40C60N and 20C80N induce unconsciousness faster and with less variability compared to the traditional 40C90C method. This research contributes to enhancing the welfare of broiler chickens during the stunning process and provides valuable insights into alternative stunning methods.

The main question addressed by the research is whether gas mixtures of carbon dioxide (CO2) associated with nitrogen (N2) can be effective alternatives to traditional CO2 stunning methods in broiler chickens during slaughter, with a focus on improving animal welfare.

The topic is both relevant and original in the field, as it addresses the crucial issue of animal welfare during the stunning process in the poultry industry. It explores alternative methods to reduce aversion and improve the efficiency of stunning, which is of significant importance for the ethical treatment of animals in the food production chain.

This research adds valuable insights to the subject area by providing experimental evidence of the effectiveness of gas mixtures in stunning broiler chickens. While there is existing literature on stunning methods, this study specifically evaluates the welfare and efficiency aspects of using different gas mixtures, contributing to a better understanding of alternative stunning techniques.

However, I would like to highlight several areas that could benefit from improvement:

I suggest rewriting the abstract to enhance its readability. Additionally, including some key details about the materials and methods used in the research would be valuable.

While the introduction is comprehensive, I recommend introducing a brief initial paragraph addressing the general welfare of poultry. For instance, I suggest considering reading and citing the following papers 0.3390/ani12182307;  10.3390/ani13132074.

I also recommend expanding the introduction to include a more comprehensive assessment of welfare, not only during slaughter but also during the preceding transport period, for poultry and other species, such as pigs, as discussed in articles 10.3390/ani10122386 and 10.3390/ani10060945.

The statistical methods described in the manuscript may benefit from further elaboration and clarity. It would be helpful to provide more details on the specific statistical tests employed, their rationale, and how they align with the research questions.

Could you please clarify whether you conducted tests for normality and homogeneity on your data before proceeding with the statistical analysis? It's crucial to ensure that the assumptions underlying your chosen statistical methods are met. I recommend referring to the guidelines outlined in [proposed reference, e.g., 10.1080/1828051X.2020.1827990] for conducting such tests to maintain the rigor and reliability of your analysis.

Expanding the discussion section to incorporate study limitations and practical applications would enhance the overall quality of the paper.

The conclusions are consistent with the evidence and arguments presented in the study. The research demonstrates that gas mixtures 40C60N and 20C80N induce unconsciousness faster and with less variability compared to the traditional 40C90C method, aligning with the aim of improving animal welfare during stunning.

The references seem suitable and pertinent to the research topic, offering valuable context and background information regarding stunning methods and animal welfare in the poultry industry. Nevertheless, I recommend incorporating more recent and broader perspective papers into this manuscript.

Reviewer 4 Report

Comments and Suggestions for Authors

Congratulations to the authors for contributing valuable evidence to this very relevant topic. Below you will find some suggestions that hopefully will be helpful in order to improve the manuscript.

Ll 34-35: I recommend the authors changing “provided the highest welfare” toproved the least aversive or the three treatments”. This wording would work best in the light of the results – and therefore in order to illustrate that all three treatments are still problematic in terms of animal welfare.

Ll 44-45: I suggest the authors change “it is essential to induce an effective stunning” to “it is essential to induce the fastest effective stunning possible”.

Ll 84-87: I find the sentence “lose consciousness by anoxia [3] which it (sic) is not perceived by the birds [12] due to lack of chemoreceptors on air ways to inert gases. In this sense, inhalation of inert gases is expected not to cause aversive reactions” problematic. The authors are right in that the chickens do not have tools for perceiving inert gases. These animals, however, can certainly perceive the effects of hypoxic/anoxic conditions and suffer as a consequence of them. The authors must reflect this in the paragraph above. 

Ll 98-99: I suggest changing “and therefore, behaviours that occur before LOC could indicate aversion while during LOC could be related to convulsions.” to “and therefore discern between aversive behaviours that occur before LOC and likely unconscious convulsions.” 

Ll 131-137: Please explain the reasons for the reported number of subjects per experimental treatment if the total number of subjects was 293 and the birds “were allocated in equal number to one of the three experimental treatments”.

Table 1: Day 3/1500 to 1900h Session (20C60N) is probably a typo. Please correct (20C80N). Also, I would appreciate it if the authors could provide the rationale for an uneven experimental design with 3 repetitions for the 40C90C treatment and two for the 40C60N and 20C60N treatments.

Round 2

Reviewer 2 Report

Comments and Suggestions for Authors

The manuscript was excellently modified and the authors addressed all my comments and questions appropriately. I have some suggestions for small edits, for which I do not expect an author response.

Simple summary: it is unclear to me where results are mentioned. I recommend shortening the problem description to add at least 1 sentence summarizing the study outcome.

Overall, the abstract was appropriately edited and my concerns were addressed. Thank you for clarifying this section!

L31: sentence needs rephrasing for this part: “but their … motionless”

L182-183: thank you for adding the manufacturer detail. I am wondering if sounding line is the appropriate terminology? Would an ‘analyzer’ or just ‘sensor’ be clearer?

L185: if I understand the content correctly, consider rephrasing to: “In the first phase of 40C90C, the concentration of CO2 varied, but was close to 40% and never exceeded that level.”

L238-247: thank you for clarifying these measures, this section is helpful.

Figure 3-4: I appreciate these figures as they nicely visualize the behavioral responses of the birds, and presenting the individual bird experience. The occurrences of head shaking and other point-behaviors are difficult to spot, so the authors may want to consider presenting those differently (can the ‘spots’ be enlarged?).

L437: replace ‘give figure’ with figure reference if that is what was meant.

The discussion was greatly improved and much clearer for readers without EEG expertise (me).

L788: rephrase to “the smaller proportion…”

Comments on the Quality of English Language

Some minimal English improvements can be made, see earlier comments in previous section. 

Author Response

Dear Reviewer,

We are deeply thankful for the time and dedication you have shown in reviewing our paper. I think your feedback has significantly enhanced the quality of our work. A pity not to know who you are...

Thank you again.

PS: Please, see the atachment.

Reviewer 3 Report

Comments and Suggestions for Authors

Dear Authors,

I wanted to extend my heartfelt congratulations to you and your team for the outstanding job you've done in revising your paper. I am genuinely impressed by the way you have meticulously incorporated the suggested revisions. Your commitment to improving the article's quality is evident, and I must say that the final result is nothing short of exceptional. The transformation from the initial draft to the current version is remarkable and a testament to your dedication to excellence.

Author Response

The authors would like to thank you for your feedback!